# Investigating the effectiveness of care delivery at an acute geriatric community hospital for older adults in the Netherlands: a protocol for a prospective controlled observational study

Marthe E Ribbink [ID] ,[1] Janet L Macneil-Vroomen,[1,2] Rosanne van Seben,[1] Irène Oudejans,[1] Bianca M Buurman[1,3] on behalf of the AGCH study group

[1]Department of Internal Medicine, Section of Geriatric Medicine, Amsterdam University Medical Centres, Amsterdam, Noord-Holland, Netherlands
[2]Department of Internal Medicine, Section of Geriatrics, Yale School of Medicine, New Haven, Connecticut, USA
[3]ACHIEVE-Centre of Applied Research, Faculty of Health, Amsterdam University of Applied Sciences, Amsterdam, Noord-Holland, Netherlands

**Correspondence to**
Marthe E Ribbink;
m.e.ribbink@amsterdamumc.nl

## ABSTRACT

**Introduction** Hospital admission in older adults with multiple chronic conditions is associated with unwanted outcomes like readmission, institutionalisation, functional decline and mortality. Providing acute care in the community and integrating effective components of care models might lead to a reduction in negative outcomes. Recently, the first geriatrician-led Acute Geriatric Community Hospital (AGCH) was introduced in the Netherlands. Care at the AGCH is focused on the treatment of acute diseases, comprehensive geriatric assessment, setting patient-led goals, early rehabilitation and streamlined transitions of care.

**Methods and analysis** This prospective cohort study will investigate the effectiveness of care delivery at the AGCH on patient outcomes by comparing AGCH patients to two historic cohorts of hospitalised patients. Propensity score matching will correct for potential population differences. The primary outcome is the 3-month unplanned readmission rate. Secondary outcomes include functional decline, institutionalisation, healthcare utilisation, occurrence of delirium or falls, health-related quality of life, mortality and patient satisfaction. Measurements will be conducted at admission, discharge and 1, 3 and 6 months after discharge. Furthermore, an economic evaluation and qualitative process evaluation to assess facilitators and barriers to implementation are planned.

**Ethics and dissemination** The study will be conducted according to the Declaration of Helsinki. The Medical Ethics Research Committee confirmed that the Medical Research Involving Human Subjects Act did not apply to this research project and official approval was not required. The findings of this study will be disseminated through public lectures, scientific conferences and journal publications. Furthermore, the findings of this study will aid in the implementation and financing of this concept (inter)nationally.

**Trial registration number** NL7896; Pre-results.

## Strengths and limitations of this study

► This study will be the first to evaluate an acute geriatric community hospital in the Netherlands on both patient-reported and economic outcomes.
► Patients, informal caregivers and professionals were involved in the design and implementation of the Acute Geriatric Community Hospital.
► A process evaluation is planned to describe the experience of various stakeholders with this new concept and reveal barriers and facilitators to its implementation.
► A limitation of this study is the use of two historic cohorts as the control population, which may result in baseline differences between the control and intervention population.

Inpatient services are mostly consumed by those over the age of 65.[1 2] The Netherlands, like many other countries, recently (2015) implemented stay-at-home policies leading to an increase in frail older persons living longer in the community.[3] These reforms juxtaposed with an increased ageing population contribute to increased acute care utilisation.[4] There has been a 19% increase in emergency department (ED) visits by Dutch older adults based on data from 2015 versus 2017.[5 6]

Many older adults come to the hospital with complex and atypical health problems.[5 7] When older persons are subsequently hospitalised, health outcomes are known to be poor,[8] particularly in patients with geriatric syndromes such as cognitive impairment or mobility impairment.[9 10] For example, previous research showed that 30% of older persons gained new disabilities and 20% were readmitted within 30 days postdischarge.[11 12] Hospitalisation itself may contribute to these

## INTRODUCTION
### Background
Throughout the western world, there is an increase in older adults requiring acute care.

poor outcomes, as hospitalised older adults often have reduced mobility because they are bedbound for approximately 20 hours a day.[13 14] Low physical activity, in combination with poor nourishment and increased caloric demand due to acute illness, can lead to the loss of muscle mass and may contribute to the development of new disabilities, particularly in frail patients.[15 16] Together with the noise in a hospital environment and the different personnel rotating between patient rooms, this contributes to sensory overstimulation and sleep deprivation, which may lead to confusion and the occurrence of delirium.[17–19] Not only is the patient affected during hospitalisation but the informal caregivers also find hospital admissions stressful.[20] Furthermore, previous research shows that a lack of discharge planning in the hospital can result in patients' care needs being unmet.[21] Hospital care as usual compared with discharge planning and follow-up show a higher rate of early readmissions.[22] Readmissions can further affect patients' recovery and increase healthcare costs.[23]

The complex medical needs of older persons, combined with their more dependent social situation, requires care delivery that offers guidance and support for realistic health and life goals.[24] Perhaps a 'gap' exists between what care can be provided in an acute care hospital versus what can be provided in the community (primary care). Acute hospital care is secondary care with a focus on medical treatment and diagnostics, while primary care focuses on rehabilitation, nursing care and well-being.

Several alternative strategies to hospital admission and (nurse-led) intermediate care have been developed in the past as a substitute to conventional hospitalisation.[25] Examples include (nurse-led) intermediate care and subacute geriatric care units, which are low-tech but with geriatric expertise.[26 27] In general, these types of care have comparable outcomes to hospital care as usual. Moreover, nurse-led care in the USA, observation units and hospital at home care all show a cost reduction compared with care as usual.[25 26] Until recently, the Netherlands had limited alternatives to hospitalisation for older persons who required acute care. Therefore, our research group sought to create an acute care alternative and opened the Acute Geriatric Community Care Hospital (AGCH) in July 2018, partnering with an academic hospital (Amsterdam UMC, location AMC), an insurance company (Zilveren Kruis) and a home care and nursing home agency (Cordaan). This acute geriatric care unit, which is based within an intermediate care facility, provides an alternative to conventional hospitalisation and delivers acute care closer to home.

The AGCH delivers acute care that is focused on early mobilisation and rehabilitation. Older persons with common medical problems (such as urinary tract infections, pneumonia or heart failure) and geriatric syndromes requiring hospital admission can be admitted to the AGCH. The AGCH provides a form of *intermediate* care between primary and secondary care. In the Netherlands, primary care includes general practice, community nursing and (temporary) admission to a nursing home. Secondary care includes specialist medical care and hospital admission. The care at the AGCH is supervised by a geriatrician and provided by nurses trained in geriatric care who have experience as either a hospital or community nurse. The single rooms are designed to accommodate respite for the informal caregivers. This concept of care is new to the Netherlands, and to our knowledge, there is only one comparable example in Europe: a 'subacute care unit' in intermediate care, which has been implemented in Spain.[27]

Our hypothesis is that with the provision of integrated medical and nursing care close to home, the AGCH is better suited to the needs of older adults with multiple chronic conditions and will lead to better patient health outcomes and reduced post-acute care costs. Therefore, this study is designed to compare care provided for older patients in the AGCH versus care provided in a hospital setting. Specifically, we aim to:

► Evaluate the 90-day readmission rate of patients acutely admitted to the AGCH compared with a traditional hospital (usual care). Secondary outcomes include functional decline, institutionalisation, healthcare utilisation, the occurrence of geriatric syndromes such as delirium, health-related quality of life (HRQOL), mortality and patient satisfaction;
► Assess the cost-effectiveness of the AGCH versus usual care by performing an economic evaluation from a healthcare provider and societal perspective;
► Conduct a process evaluation using interviews with key stakeholders to identify facilitators and barriers to the implementation of the AGCH.

## METHODS
### Setting
The AGCH opened in July 2018. It serves the south-eastern part of Amsterdam and its surrounding areas (an area with approximately 147 500 inhabitants).[28] The AGCH is a 23-bed facility within a skilled nursing facility. The hospital has 24 hours geriatric and nursing assistance. Physiotherapy and routine laboratory testing are available during the workweek and simple X-ray is available once a week. The population that is eligible for admission to the AGCH are patients with a combination of an acute medical problem requiring hospitalisation (eg, pneumonia, exacerbation of heart failure or a urinary tract infection) and a geriatric condition (eg, delirium, cognitive impairment, falls, or functional impairment). Additionally, patients have to be haemodynamically stable and should not require complex diagnostic testing. In general, patients will not be admitted if they have the following exclusion criteria: (1) require care that can only be provided at an intensive care unit, (2) require surgery, (3) require urgent treatments or diagnostic tests that can only be provided in-hospital (eg, endoscopy, interventional radiology), (4) do not need hospital care but require transfer to a skilled nursing facility and (5) live in another region of the Netherlands.

Patients are directly admitted to the AGCH from the ED of the Amsterdam UMC-location Academic Medical Centre (AMC) in Amsterdam, which is a 1000-bed academic hospital with approximately 30 000 ED visits yearly. After the on-call geriatrician has assessed whether the patient is eligible for AGCH admission and the patient or representative has agreed to admission, the patient is transferred to the AGCH by ambulance. Since October 2019, patients can also be transferred from the EDs of other hospitals in Amsterdam. In the future, we plan to admit patients from home or general practice offices. Patients are admitted between 8.00 am and 23.00 pm, 7 days a week. At admission, a Comprehensive Geriatric Assessment (CGA) is conducted.[29] The CGA gives an overview of all medical, functional, psychological and social problems. The CGA is discussed during multidisciplinary team meetings and used to formulate a care plan for each patient. For an overview of the admission process, the admission criteria and the components of this intervention, see figure 1.

### Study design

This study is a prospective, observational, cohort study with two historical control groups to evaluate the clinical and economic effects of the AGCH. The Strengthening the Reporting of Observational Studies in Epidemiology (STROBE) statement was used in preparing the study protocol (online supplementary appendix 1).[30] Participants will be compared with hospital controls. The participants are recruited into the study and are assessed at admission, discharge, and 1, 3 and 6 months after discharge. Recruitment for this study started in February 2019. We plan to recruit for 18–24 months. The first 3 months of data collection consisted of a piloting phase to

assess the feasibility of data collection and follow-up. In addition, a qualitative process evaluation on the facilitators and barriers to the implementation of the AGCH and patient experience will be conducted.

### Participants

Patients admitted to the AGCH are eligible for inclusion in the study. However, patients are excluded from the study if: (1) the attending physician judges that the patient is too ill to participate, for example, the patient is terminally ill, (2) the patient or legal representative does not consent to participate, or (3) the patient or legal representative does not speak or understand Dutch or English. In the case of cognitively impaired or delirious patients, patients can only be included if a legal representative consents to participation and acts as healthcare proxy. Cognitive functioning is assessed by the attending physician and confirmed by the researcher by conducting a Mini-Mental State Examination (MMSE).[31] An MMSE score of 15 or less indicates severe cognitive impairment, in which the approval of a legal representative will be sought.

### Historical control groups

We selected two completed cohort studies that were conducted by our research group as historical control groups. We expect that the patients from these cohorts have similar admission diagnoses as those who can be admitted to the AGCH, namely, diagnoses that are ambulatory care sensitive conditions such as infections and exacerbations of chronic obstructive pulmonary disease (COPD) or heart failure.[32] Patients in these two cohorts were admitted to internal medicine, cardiology, pulmonology and geriatrics departments. These

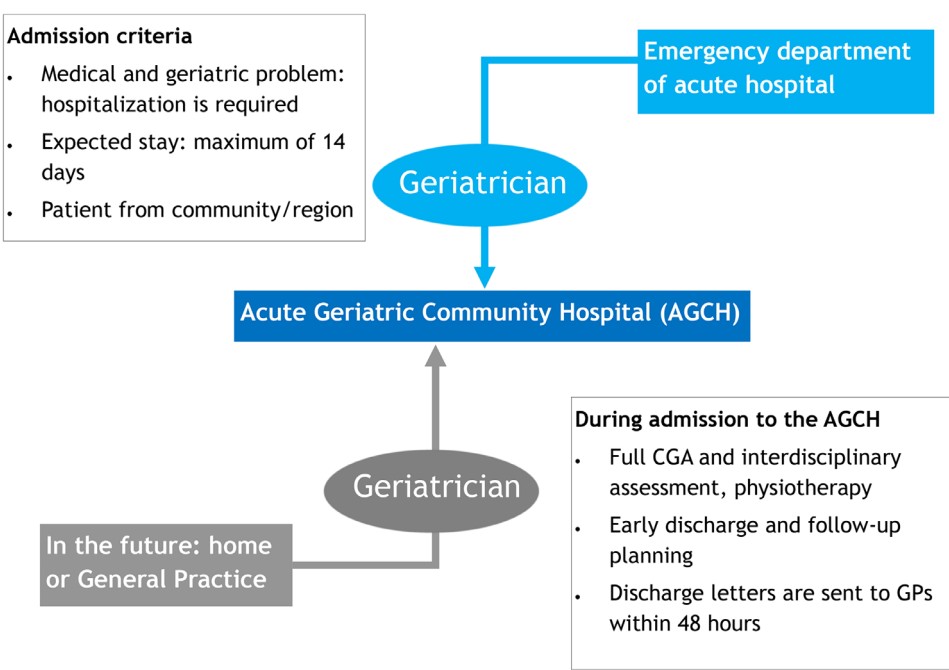

**Figure 1** Patient admission process and criteria, components of the AGCH intervention and goals. CGA, Comprehensive Geriatric Assessment[29]; GP, General practitioner.

departments admit patients with diagnoses similar to those that can be admitted to the AGCH. In addition, we have selected these cohorts as control groups as the patients come from the same area as the studied population admitted to the AGCH, that is, the greater Amsterdam area. The first control group from the Transitional Care Bridge Study consists of 674 patients who were recruited between September 2010 and March 2014.[33] Participants were patients of 65 years and older hospitalised for at least 48 hours. Proxy consent was provided for participants suffering from severe cognitive impairment (MMSE ≤15). They participated in a negative randomised controlled trial that assessed the effectiveness of a nurse-led transitional care programme in preventing functional decline.[33] The second control group from Hospital-Associated Disability and impact on daily Life study (Hospital-ADL study) consists of 401 patients who were recruited between October 2015 and June 2017.[10] These participants were enrolled in a prospective cohort studying the trajectory of functional decline in older hospitalised adults. Participants were aged 70 years and older and were hospitalised for at least 48 hours. Patients suffering from severe cognitive impairment (MMSE ≤15) and delirium were excluded from participation. For the detailed methodology and inclusion criteria of the two control cohorts, please refer to the study protocols and papers of these studies.[10 33–35]

### Patient and public involvement

Older persons living in Amsterdam were involved in the design of the AGCH concept. No patients were involved in the design of this study.

### Outcomes

The primary outcome measure is the 3-month unplanned readmission rate to the AGCH or hospital.

Secondary outcomes measured at 1, 3 and 6 months will include:

1. Activities of Daily Living (ADL)-functioning, as defined by the Katz-ADL scale.[36]
2. Healthcare utilisation, including institutionalisation in a long-term care facility.
3. Occurrence of delirium and/or falls.
4. HRQOL.[37]
5. All-cause mortality.
6. Satisfaction of the patients and primary caregivers with the provided care.

### Data collection

Eligible patients and/or legal representatives will be contacted and informed about the study procedures after which written informed consent is obtained. Inclusion and interviewing of patients is conducted by an onsite researcher. Routine data on functioning and risk assessments are collected by a trained registered nurse and physiotherapist as part of the CGA for each patient.[38] Table 1 gives an overview of measurement of the primary and secondary outcomes over time. These measurements were chosen based on the assessments and data collected

from the two historic control groups. The online supplementary table provides an overview of the content and timing of measurements in the AGCH-group compared with the two historic control groups. Measurements during admission are at H1, which is within 48 hours after admission, and H2, which is within 48 hours before discharge. Follow-up is completed by telephone at 1, 3 and 6 months after discharge (P1, P3 and P6).

Data collection includes the following.

### Medical and demographical data
#### Sociodemographic data

These will include age, gender, highest level of education, ethnicity, marital status and living arrangement.

#### Data on admission

Time spent at the ED, admission diagnosis and date and time of admission.

#### Chronic conditions

The number and severity of chronic conditions will be assessed using the Charlson Comorbidity Index.[39] This index is commonly used to indicate the risk of mortality; each condition is scored 1, 2, 3 or six points, with a higher total number of points indicating a greater risk of death.

#### Polypharmacy

Polypharmacy will be assessed by counting the number of individual drugs that are chronically prescribed to a participant, in which a number of five or more drugs is considered polypharmacy.

#### Mortality

This will be assessed during follow-up, either from the patients' electronic files or from general practice registries.

### Cognitive functioning
#### Cognitive impairment

This is assessed by reviewing the score of the MMSE that is performed within 48 hours of admission. The MMSE includes 23 items (total score 0–30) that screen for cognitive impairment. A score of 23 or less is defined as possible cognitive impairment.[31] When a patient is delirious on inclusion, the MMSE is not conducted.

#### Delirium

The Confusion Assessment Method (CAM), four-item short version, is used to assess the presence and duration of delirium.[40] The CAM is widely used by physicians and nurse practitioners to diagnose delirium (sensitivity of 53%–90% and specificity of 84%–100%).[41] The CAM is filled out within 24 hours of admission. Moreover, the risk on developing delirium is assessed using the Dutch Safety Management Programme (*Veiligheidsmanagement-systeem (VMS)*) criteria for risk of delirium.[42] Nurse practitioners will score the CAM daily from day 1 till day 3 of admission; if there are signs of possible delirium at day 3, these measurements are continued until the symptoms

**Table 1** Overview of the content and description of the (outcome) measurements and timing of the measurements at the acute geriatric community hospital (AGCH)

| | Description and/or instrument | H1 | H2 | P1 | P3 | P6 |
|---|---|---|---|---|---|---|
| **1. Medical and demographical data** | | | | | | |
| Sociodemographic data | Date of birth, age at admission, sex, level of education, living conditions, marital status | R | | | | |
| Data on admission | Time spent at the ED, admission diagnosis, date and time of admission | R | | | | |
| Chronic conditions | Charlson Comorbidity Index[39] | R | | | | |
| Polypharmacy | Number of drugs | R | | | | |
| Mortality | Date of death | | R | R | R | R |
| **2. Cognitive functioning** | | | | | | |
| Cognitive impairment | Mini-Mental State Examination[31] | R | | | | |
| Delirium | Safety management system patient screening (VMS)[42] Confusion Assessment Method[40] Delirium Observation Scale[43] | N/D | N/D | | | |
| **3. Psychosocial functioning and quality of life** | | | | | | |
| Apathy | Geriatric Depression Scale[44] | N | R | R | R | R |
| Social network and informal care | Presence and frequency of informal care | R | | R | R | R |
| Quality of life and health status | EQ-5D[37] | R | | R | R | R |
| **4. Physical functioning** | | | | | | |
| Identifying at-risk-patients | ISAR-HP—Identifying Seniors at Risk score[45] | N | | | | |
| Functional status | Activities of Daily Living (ADL) modified Katz-ADL score[36] | N | | | | |
| (Im)mobility | Using a walking aid, information from the Katz-ADL questions on exercise | N | | | | |
| Handgrip strength | Jamar[49] | P | | | | |
| Gait speed | Short Physical Performance Battery[50] | P | | | | |
| Falling | Fall history Falls in the AGCH Numeric Rating Scale (NRS) on the fear of falling[35] | N N | R R | R R | R R | R R |
| Pain | NRS on pain[51] | N | R | R | R | R |
| Fatigue | NRS on fatigue[52] | N | R | R | R | R |
| Nutrition | Short Nutritional Assessment Questionnaire[53] | N | | | | |
| **5. Healthcare utilisation and satisfaction with care** | | | | | | |
| Medical care during admission | Diagnostics performed in the AGCH Readmission to university hospital Length of stay at the AGCH | | | R | | |
| Hospital readmission | Readmission rate to the hospital or AGCH | | R | R | R | R |
| Healthcare utilisation | Home care, medical specialist care, temporary institutional care, primary care | R | | R | R | R |
| Satisfaction with care | 8-question questionnaire[54] | | R | (R)* | | |

*In case the assessment was missed at H2.

AGCH, Acute Geriatric Community Care Hospital; D, Doctor/attending physician; ED, emergency department; EQ-5D, EuroQoL-5D; H1, at admission; H2, at discharge; ISAR-HP, Identification of Seniors at Risk-Hospitalized Patients; N, nurse; P1, one month after discharge; P3, three months after discharge; P6, six months after discharge; P, physiotherapist; R, researcher/research nurse.

are resolved. In addition, during the first 3 days of admission, the Delirium Observation Screening Scale is scored during each nursing shift and is continued when there is a clinical suspicion of delirium.[43]

## Psychosocial functioning and quality of life
### Apathy
We use three items of the *Geriatric Depression Scale* (GDS-15) to assess apathy (sensitivity of 69% and specificity of 85 %). These items include the following questions: (1) 'Do you prefer to stay at home, rather than going out and doing new things', (2) 'Have you dropped many of your activities and interests?' and (3) 'Do you feel full of energy'. A score of >2 points is classified as 'apathy present'.[44]

### Social network and informal care
Participants are asked if they receive informal care, how many hours a week, what type of care (housekeeping and/ or personal care) and from which persons (partners, children, other family members or neighbours/volunteers).

### Health-related quality of life
This will be measured by the EuroQoL-5D (EQ-5D). The EQ-5D is a broadly used and validated instrument for measuring generic HRQOL.[37]

## Physical functioning
### Risk of functional decline
Patients are assessed for risk of functional decline using the Identification of Seniors at Risk-Hospitalised Patients tool; scores of two and up indicate an increased risk for functional decline.[45]

### Functioning level
The 15-item modified Katz-ADL score is used to measure ADL functioning. This includes statements about independence in performing basic ADL and in instrumental ADL (IADL).[46 47] We measure the Katz-ADL both currently (at admission), as well as 2 weeks before admission, reflecting pre-morbid level of functioning. The Katz-ADL is also measured during follow-up.

### (Im)mobility
Mobility is assessed by reviewing three questions that are in the admission assessment regarding: (1) the use of a walking aid, (2) being able to walk outside of the house for 5 min (2 weeks before and currently) and (3) the performance and frequency of physical activity.[48]

### Handgrip strength
Measure muscle weakness is measured by physiotherapists in all admitted patients using the maximum handgrip strength (Jamar).[49]

### Gait speed
Gait speed is measured as part of the Short Physical Performance Battery, which is part of the physiotherapists' admission assessment.[50]

### Falls
Fall history is assessed by asking about the number of falls in the past 6 months.[42] During the discharge assessment, the occurrence of falls in the AGCH and the consequences of falls (indication for prolonged stay, diagnostics or injury) are recorded.

### Fear of falling
*The* Numeric Rating Scale (NRS, score 0–10) is used to assess the fear of falling; 0 indicates no fear of falling, and 10 indicates the greatest fear of falling possible.[35]

### Pain
The standard clinical measure for pain is the NRS, ranging from 0 to 10, in which a score of 0 represents no pain and 10 represents the worst possible pain.[51]

### Fatigue
An NRS from 0 to 10 is used, with 0 indicating no fatigue and 10 indicating the greatest fatigue ever felt by the participant.[52]

### Sleep
Participants are asked if they have had difficulties with sleeping in the past month and whether participants have used sleep medication.

### Nutrition
We will use the Short Nutritional Assessment Questionnaire (SNAQ) to identify patients with malnourishment. The SNAQ consists of three questions concerning weight loss, appetite and drink/tube nutrition, resulting in a score ranging from 0 to 5. Scores of 0 and 1 are defined as 'no malnutrition', 2 as 'moderate malnutrition' and 3 or more as 'severe malnutrition'.[53]

## Healthcare utilisation and satisfaction with care
### Medical care during admission and the process of discharge
The following items are collected from patients' electronic health records: the diagnostics performed in the AGCH, revisits to the hospital, admissions to the hospital, length of stay at the AGCH, discharge destination and time needed to send medical handovers to the general practitioner.

### Hospital readmission
This outcome will be assessed during follow-up. Follow-up will consist of three telephone interviews at 1, 3 and 6 months after discharge. Readmission will be both assessed during the follow-up interviews and by checking care data from an aggregated database of expense claims from various healthcare insurers. Data that will be collected are as follows: number of readmissions, total days of readmission, reasons for readmission and whether the readmission was planned or unplanned.

### ED visits
ED visits will be assessed during follow-up and checked in the insurance data. We will record the number of separate ED visits.

### Outpatient hospital visits

We will ask patients if there have been any outpatient visits in the past month(s), and if so, how many.

### Consultations by general practitioners

We will ask patients if, and how many times, they have consulted with their general practitioner (both during the day and during out-of-office hours).

### Consultations by physiotherapists or dieticians

We will ask patients if, and how many times, they have consulted with a physiotherapist or dietician in the past month(s).

### Home care

This includes questions on the frequency of home care, including housekeeping, personal care and nursing care. We will also include hours of informal care provided by family members or friends.

### Temporary admission to a nursing home

This includes days of (temporary) admission to a skilled nursing facility or rehabilitation facility.

### Permanent institutionalisation

This concerns long-term admission to a skilled nursing facility and the date of admission to this facility.

### Patient satisfaction with care

Patients or informal caregivers are asked to fill out an eight-question questionnaire regarding their satisfaction with the care that they received. Questions are answered on a five-point Likert scale.[54]

### Sample size calculation

In the Hospital-ADL study, 34% of participants experienced a readmission at 90 days.[35] Assuming that 26% of patients admitted to the AGCH will experience a 90-day readmission, data from 515 patients at the AGCH will yield 80% power to detect an absolute difference of 8% in the readmission rate (which is a 25% reduction in the relative risk) using a two-sided test with an alpha of 0.05.[55] As we expect 10% loss to follow-up, we aim to include a total of 567 (=515*1.10) patients from the AGCH.

### Planned statistical analyses

The complete participant flow diagram will show a summary of admissions and study recruitment at the AGCH and will provide study discontinuation rates at 1, 3 and 6-months follow-up.[30] We will describe the demographic, clinical and prognostic characteristics of the study participants at baseline. The number of participants with missing data will be collected and described alongside our variables to check for the pattern of missingness. Inversely weighted propensity scores will be used to control for any imbalances between the treatment groups.[56] Propensity scores will be calculated using generalised booted methods. Balance and overlap of propensity score distribution will be assessed. Propensity score weights for the estimation of the average treatment effect will be created using all covariates where groups differed at baseline or that were associated with the 90-day readmission rate. As this is a repeated measures design, we will assume equal weighting for all measurements.[57]

All hypotheses will be tested using a two-tailed significance level of 0.05. All secondary outcomes will be adjusted for multiple testing using a Hochberg method.[58 59] Descriptive analyses will be performed to examine the participants' characteristics. Differences in changes over time in outcomes will be compared between groups using multilevel models. All models will include a main effect of treatment group, a linear term for time and an interaction between time and treatment group. Models will be checked with residual and appropriate goodness-of-fit statistics.

### Economic evaluation

A healthcare and societal perspective is planned for the economic evaluation. The evaluation from the healthcare perspective will only include direct medical costs accrued in the 6 months after the admission to the AGCH. Direct medical costs will only include costs that are funded through the Dutch healthcare system. The evaluation from a societal perspective will include an estimation of the costs of informal care. Costs will be based on the reference prices found in the Dutch Manual for Costing studies and will be set for the final year of data collection (2020 or 2021). According to this guideline, costs will be discounted at 4% and quality adjusted life years (QALY) will be discounted at 1.5%.[60] Propensity scores will also be used in the economic evaluation. Missing data will be imputed using multiple imputation chained equations, if necessary, for the cost and effect data. We plan to use generalised linear regression models with a gamma distribution and an identity link to account for the right skew of the cost data. A generalised linear regression model will be used to estimate the incremental effect in QALY adjusted for baseline utility estimates with a Gaussian distribution and identify link.[61] Incremental cost-effectiveness ratios will be calculated using the pooled cost and effect estimates. Bootstrapped cost-effect pairs will be plotted on a cost-effectiveness plane and used to estimate cost-effectiveness acceptability curves.[62]

### Process evaluation and patient experience

We plan to use a qualitative study design to describe the barriers and facilitators to implementation of the AGCH concept and describe the experiences of the patients and healthcare professionals with the AGCH. We will conduct semi-structured interviews with various stakeholders, such as geriatricians, nurses, physiotherapists and hospital administrators. These interviews will concern the implementation of the AGCH concept. In addition, semi-structured interviews with patients and informal caregivers will be conducted in order to describe the patient experience and satisfaction with this new form of care. A representative sample of patients and/or caregivers who participate in the prospective cohort study will

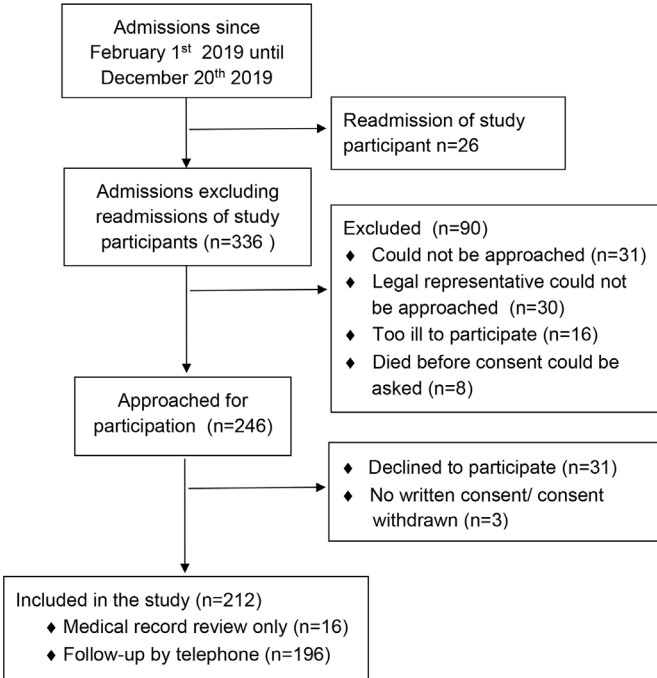

**Figure 2** Diagram of patient participation between February 1st and December 20th, 2019.

be approached and invited to be interviewed shortly after discharge from the AGCH. Stakeholders and healthcare professionals will be selected by a researcher and will be invited for an interview to discuss their experiences and opinions on the AGCH. Interviews will be typed verbatim and analysed independently by two researchers using thematic analyses.[63] In our analysis of the barriers and facilitators to implementation, we will describe these factors at three different levels: micro (healthcare professionals), meso (care organisations) and macro (legal and financial framework).[64] The findings will be summarised in matrices with the facilitators and barriers at these three different levels and can be used to develop a guideline for implementation of the AGCH elsewhere.[65]

## PRELIMINARY RESULTS

Between February 1st and December 20th, 2019, there were 362 consecutive admissions to the AGCH. Of these admissions, 26 were readmissions of patients who were already study participants. Of the remaining 336 admissions, 90 were by patients who did not meet the inclusion criteria. The remaining 246 patients or legal representatives and healthcare-proxy were approached for participation; 212 consented to participation (figure 2). The healthcare-proxy provided informed consent in 62 (29.2%) of cases. Sixteen patients did not consent to follow-up by telephone but did consent to medical record review. The total study sample as of December 20th, 2019, consisted of 212 participants at baseline. Table 2 displays the baseline characteristics of this group. Participants had a mean age (SD) of 81.8 (8.4) years and 47.6% were male. Most participants were living independently before

| Table 2 Baseline characteristics of the study participants | |
|---|---|
| **Variable** | **N=212** |
| Age in years, mean (SD) | 81.8 (8.4) |
| Male, N (%) | 101 (47.6) |
| Living arrangements before admission, N (%) | |
| Independent | 172 (81.1) |
| Assisted living/senior residence | 31 (14.6) |
| Nursing home/other | 9 (4.2) |
| Marital status, N (%) | |
| Widow/widower | 94 (44.5) |
| Married or living together | 71 (33.6) |
| Single or divorced | 46 (21.8) |
| Education, N (%) | |
| Primary school | 36 (18.7) |
| Elementary technical/domestic science school | 41 (21.2) |
| Secondary vocational education | 65 (33.7) |
| Higher level high school/third-level education | 51 (26.4) |
| Born in the Netherlands, N (%) | 158 (76.0) |
| Katz-ADL (6-item) score* upon admission, median (IQR) | 3.0 (1.0-5.0) |
| MMSE score†, mean (SD) | 23.7 (4.7) |
| Polypharmacy‡, N (%) | 159 (75.0) |
| Hospitalisation in past 6 months, N (%) | 61 (31.1) |
| Charlson Comorbidity Index§, mean (SD) | 2.8 (2.0) |
| Primary admission diagnosis, N (%) | |
| Infectious diseases | 60 (28.3) |
| Respiratory (including pneumonia) | 54 (25.5) |
| Gastrointestinal | 9 (4.2) |
| Cardiovascular | 20 (9.4) |
| Neurological | 16 (7.5) |
| Other (eg, falls, delirium, sudden unexplained functional decline) | 53 (25.1) |

*Score ranging from 0 to 6, with a higher score indicating more dependence in activities of daily living.[36]
†Score ranging from 0 to 30, with a score of ≤23 indicating possible cognitive impairment.[31]
‡Use of five drugs or more.
§Ranging from 0 to 31, with a higher score indicating more severe comorbidity.[39]
ADL, activities of daily living; MMSE, Mini-Mental State Examination.

admission (81.1%). The most frequent admission diagnoses were infectious diseases (28.3%, mostly urinary tract infections), respiratory-related diseases (25.5%, of which half were pneumonia) and other (geriatric) diagnoses such as falls, delirium or sudden unexplained functional decline (25.1%). The main cardiovascular (9.4%) admission diagnosis was exacerbation of heart failure. The median (IQR) length of stay was 8.0 days (5.0–12.0)

and 83.7% of patients were discharged to their original living situation.

## DISCUSSION

The complex acute medical needs of older patients require the delivery of specialised geriatric care. The traditional hospital environment may however not support recovery and maintaining independence. The AGCH aims to deliver care that focuses on medical treatment, early rehabilitation and proper transitions of care for older adults with multiple chronic conditions.[29 66] The AGCH is unique in the Netherlands in its aim to combine multiple evidenced-based components of care for frail older persons at an alternative location for hospital care. The proposed research will provide insight into the clinical and economic effectiveness of care delivered at the AGCH, compared with hospital care.

Our preliminary results show that data collection at the AGCH is feasible and we expect to recruit enough patients to evaluate the primary outcome. There are also limitations to the design of this study. It is a non-randomised study and historic cohorts are used as control groups. Therefore, baseline differences between the intervention and control groups may hamper the matching between the groups. Additionally, the data from the historic cohorts were not collected in the same time period as the AGCH cohort. This is a limitation as work processes in hospitals may have changed over the years, which could influence our results. However, the two control populations do represent a geriatric population that was admitted for exacerbations of chronic conditions and acute illnesses that frequently occur in older persons. The strengths of the study are the involvement of patients and informal caregivers in the design of the concept of the AGCH. Moreover, a process evaluation will address the barriers and facilitators to implementation of the AGCH in the Dutch Healthcare system. In short, this research will provide valuable insights into the implementation of this concept of care in other regions of the Netherlands and abroad.

## ETHICS AND DISSEMINATION

Based on the study protocol, the Ethics Committee (METC) of the Amsterdam University Medical Centre waived the obligation for the study to undergo formal ethical approval as is described under Dutch law in the Medical Research in Humans Act, January 2019 (ref W17_474 # 19.001). As this is a prospective study and pseudonymised data is used, written informed consent was obtained from the participants prior to participation. This is in line with current European legislation under the General Data Protection Regulation (GDPR).

This study will be carried out in accordance with the Declaration of Helsinki and current ethical requirements. The outcomes of this study will be reported according to the STROBE guidelines for cohort studies.[30] This study will evaluate both the effectiveness of this type of care delivery and the costs that are involved, allowing for this concept to be implemented elsewhere. The findings of this study will be published in peer-reviewed journals.

**Acknowledgements** The authors thank all the (care) professionals from the Amsterdam University Medical Centres, Cordaan and Zilveren Kruis who have worked on the development of the AGCH. Thank you for your time, advice and cooperation. We would also like to thank the members of the AGCH study group, which are the clinicians who work at the Geriatrics Department of the Amsterdam University Medical Centres and who support the data collection at the AGCH.

**Collaborators** The AGCH study group: R Franssen; W J Frenkel; M J Henstra; M A van Maanen; J L Parlevliet; E P van Poelgeest; M N Resodikromo; K J Kaland; N van der Velde; M E Visser; H C Willems.

**Contributors** MER, JLM-V and RvS contributed to the design of the protocol and drafting of the manuscript. IO and BMB made substantial contributions to the design and clinical aspects of the of the protocol. BMB conceived the study and wrote funding applications. All authors critically revised the manuscript and approved the final version of this manuscript.

**Funding** This research received funding through ZonMw, the Netherlands Organisation for Health Research and Development, project number 808393598041. The care provided at the AGCH (WijkKliniek) is provided in a partnership between Cordaan, a community and home-care organisation, and the Amsterdam University Medical Centres, location Academic Medical Centre. The AGCH (WijkKliniek) is financially supported by Zilveren Kruis, a health insurance company.

**Competing interests** None declared.

**Patient and public involvement** Patients and/or the public were not involved in the design, or conduct, or reporting, or dissemination plans of this research.

**Patient consent for publication** Not required.

**Provenance and peer review** Not commissioned; externally peer reviewed.

**ORCID iD**
Marthe E Ribbink http://orcid.org/0000-0002-5314-0520

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
