## [Reviewer comments · BMJ Open]

ARTICLE DETAILS

TITLE (PROVISIONAL)	Investigating the effectiveness of care delivery at an acute geriatric community hospital for older adults in the Netherlands: a protocol for a prospective controlled observational study
AUTHORS	Ribbink, Marthe Elisabeth; Macneil-Vroomen, Janet L; van Seben, Rosanne; Oudejans, Irène; Buurman, Bianca M.

VERSION 1 – REVIEW

REVIEWER	Miharu Nakanishi Tokyo Metropolitan Institute of Medical Science, Japan
REVIEW RETURNED	20-Sep-2019

GENERAL COMMENTS	The protocol is well-documented in general. There are two minor concerns that would be addressed: 1. If each historical cohort contains all measures mentioned in outcomes and data collection; and2. A unique point of intervention that is not duplicated by other approaches.
--

REVIEWER	Tammy Hshieh Brigham and Women's Hospital, United States of America
REVIEW RETURNED	10-Dec-2019

GENERAL COMMENTS	The authors present the study protocol for an important project – and an ambitious undertaking – evaluating an acute geriatric community hospital for older adults. Overall, the scope of this study may be too ambitious and difficult to accomplish – there are standard metric outcomes (delirium incidence, readmission, etc) in addition to patient/caregiver outcomes and economic outcomes. The study design appears robust and well thought out. The introduction is well written. The caveat of course is that the real world is messier and one wonders about some of the logistics. For example, how many patients and their families are amenable to being transferred from an acute care hospital emergency room to an intermediate care facility like this? How many patients undergo all the assessment delineated – including quality of life, nutrition, caregiver burden assessment? Given that this acute geriatric community hospital has already been in place since 2018, some preliminary/baseline data would be informative. This reviewer was left wondering how healthcare utilization will be analyzed – the authors state only direct costs would be included. Additional information about this would be important and helpful. For the analysis, this reviewer wonders a bit about the historical control groups. Why was the Transitional Care Bridge Study
---

	patient population and the Hospital-ADL patient population chosen to be historical control groups for this current study? Are they truly representative of the general geriatric population or are these patients self-selected or skewed because they are already involved in research studies? This is not a limitation that cannot be overcome, but the authors should discuss this in some more detail in the Discussion. And the justification of choosing these two groups as historical controls could be more fully explained in the methodology. Other Major Points:  • Regarding the time points, did the authors consider expanding the timeframe to 6-12 months after hospitalization? There may be valuable information at the 12 month (or at least 9 month) time point. • Pg. 15: Very robust assessment of patients! This reviewer wonders if all this data collection is feasible in the real world setting (it is a geriatrician's dream wealth of data!) • Similar to the point above re: Pg. 15: On pg. 19, Line 399 the discussion of Process Evaluation – adherence, barriers and facilitators to implementation could be fleshed out in more detail. • I do not find Box 1 on Pg. 25 to be very informative. A figure would be more useful, perhaps showing how patients are recruited in ED → go to the Acute Geriatric Hospital → warm handoff with community nurses → follow up with general practitioners in the community Minor points:  • Pg. 8, Line 124: Change “specialist” to “acute” or “tertiary” care? • Pg. 10, Line 198: Data has been collected from February 2019 through May 2019 already. Can this baseline data be included in this paper – or at least a discussion confirming feasibility of data collection? • Pg. 13, Line 264: “This will” is missing a “be”
--	--

VERSION 1 – AUTHOR RESPONSE

Response to Reviewer 1

The protocol is well-documented in general. There are two minor concerns that would be addressed:

1. If each historical cohort contains all measures mentioned in outcomes and data collection:

Author response:

Thank you for your remark. Many measures are measured in both the Transitional Care Bridge (TCB) and Hospital-ADL (H-ADL) study. However there will be some items, such as the full Short Physical Performance battery (SPPB) and the Confusion Assessment Method (CAM) that we did not collect in respectively TCB (SPPB) or H-ADL (CAM). This is also the case for the patient satisfaction questionnaire.

Please also see our answer to the comment number three made by the editor.

Please find an overview of which measures are collected in the two historical cohort below:

Supplementary table Overview of the content and description of (outcome) measurements and timing of measurements at the Acute Geriatric Community Hospital (‘WijkKliniek’) compared to the two control groups.

Grey tone= measurement in prospective cohort study. H1= at admission, H2= at discharge, P1= one month after discharge, P3 = three months after discharge, P6 = six months after discharge. T= available from *Transitional Care Bridge study(TCB)*¹
H= available from *Hospital- ADL study(H-ADL)*² *Not available from TCB or H-ADL †One baseline measurement n.a.= not applicable

	Description and/or instrument	H1	H2	P1	P3	P6
1. Medical and demographical data						
Sociodemographic data	Date of birth, age at admission, sex, level of education, living conditions, marital state	T, H				
Data on admission	Time spent at the ED*, admission diagnosis, date and time of admission	T H				
Chronic conditions	Charlson Comorbidity index ⁵	T H				
Polypharmacy	Number of drugs ⁶	T H				
Mortality	Date of death				T, H	
2. Cognitive functioning						
Cognitive impairment	Mini Mental State Exam (MMSE) ⁷	T H				
Delirium	Safety management system patient screening (VMS) ⁸	T				
	Confusion Assessment Method (CAM) ⁹	TH† T†				
	Delirium Observation Scale (DOS) ¹⁰					
3. Psychosocial functioning and quality of life						
Apathy	Geriatric Depression Scale (GDS-3) ¹¹	H	H	H	H	
Social network and informal care	Presence and frequency of informal care	T		-	-	T
Quality of life and health status	EQ-5D-3L ¹²	T H		H	H	T H
4. Physical functioning						
Identifying at-risk-patients	ISAR-HP- Identifying Seniors at Risk score ¹³	T				
Functional status	Activities of daily Living (ADL) modified Katz-ADL score ¹⁴	T H	H	H	H	T
(Im)mobility	Using walking aid, information in KATZ-15 questions on exercise	T H				
Handgrip strength	Jamar ¹⁵	T H	H			
Gait speed	Short Physical Performance Battery SPPB ¹⁶	T H	H			
Falling	Fall history	T H	-	H	H	T
	Falls in the AGCH	n/a.	n/a	na.	na.	na.
	Numeric Rating scale (NRS) fear of falling ²	H	H	H	H	-
Pain	Numeric Rating Scale (NRS) pain ¹⁷	T H	H	H	H	-
Fatigue	Numeric Rating Scale (NRS) fatigue ¹⁸	T H	H	H	H	-
Nutrition	Short Nutritional Assessment Questionnaire (SNAQ- Score) ¹⁹	T H				
5. Healthcare utilization and satisfaction with care						
Medical care during admission	Diagnostics performed in the AGCH Readmission to university hospital Length of stay at the AGCH		n/a			
Hospital readmission	Readmission rate to the hospital or AGCH			H	T H	T
Health care utilization	Home care, medical specialist care, temporary institutional care, primary care.	T		H	H	T

Satisfaction with Care	8 question questionnaire ²⁰		-	-		
--	--	---	---	--	--

References for this table

1. Charlson ME, Pompei P, Ales KL, MacKenzie CR. A new method of classifying prognostic comorbidity in longitudinal studies: development and validation. *Journal of chronic diseases*. 1987;40(5):373-383.
 2. Zorginstituut N. Veelgestelde vragen: polifarmacie <https://www.gipdatabank.nl/veelgestelde-vragen/polyfarmacie>. Published 2017. Accessed 8th of January 2019.
 3. Folstein MF, Folstein SE, McHugh PR. "Mini-mental state". A practical method for grading the cognitive state of patients for the clinician. *Journal of psychiatric research*. 1975;12(3):189-198.
 4. Heim N, van Fenema EM, Weverling-Rijnsburger AW, et al. Optimal screening for increased risk for adverse outcomes in hospitalised older adults. *Age and ageing*. 2015;44(2):239-244.
 5. Inouye SK, van Dyck CH, Alessi CA, Balkin S, Siegel AP, Horwitz RI. Clarifying confusion: the confusion assessment method. A new method for detection of delirium. *Annals of internal medicine*. 1990;113(12):941-948.
 6. Schuurmans MJ, Shorridge-Baggett LM, Duursma SA. The Delirium Observation Screening Scale: a screening instrument for delirium. *Res Theory Nurs Pract*. 2003;17(1):31-50.
 7. van der Mast RC, Vinkers DJ, Stek ML, et al. Vascular disease and apathy in old age. The Leiden 85-Plus Study. *International journal of geriatric psychiatry*. 2008;23(3):266-271.
 8. EuroQol--a new facility for the measurement of health-related quality of life. *Health policy (Amsterdam, Netherlands)*. 1990;16(3):199-208.
 9. Hoogerduijn JG, Schuurmans MJ, Duijnste MS, de Rooij SE, Grypdonck MF. A systematic review of predictors and screening instruments to identify older hospitalized patients at risk for functional decline. *Journal of clinical nursing*. 2007;16(1):46-57.
 10. Katz S. Assessing self-maintenance: activities of daily living, mobility, and instrumental activities of daily living. *Journal of the American Geriatrics Society*. 1983;31(12):721-727.
 11. Roberts HC, Denison HJ, Martin HJ, et al. A review of the measurement of grip strength in clinical and epidemiological studies: towards a standardised approach. *Age and ageing*. 2011;40(4):423-429.
 12. Guralnik JM, Simonsick EM, Ferrucci L, et al. A short physical performance battery assessing lower extremity function: association with self-reported disability and prediction of mortality and nursing home admission. *Journal of gerontology*. 1994;49(2):M85-94.
 13. Reichardt LA, Aarden JJ, van Seben R, et al. Unravelling the potential mechanisms behind hospitalization-associated disability in older patients; the Hospital-Associated Disability and impact on daily Life (Hospital-ADL) cohort study protocol. *BMC geriatrics*. 2016;16(1):59.
 14. McCaffery M, Beebe A. Pain: clinical manual for nursing practice. *CV Mosby: St Louis*. 1989.
 15. Hwang SS, Chang VT, Cogswell J, Kasimis BS. Clinical relevance of fatigue levels in cancer patients at a Veterans Administration Medical Center. *Cancer*. 2002;94(9):2481-2489.
 16. Kruizenga HM, Seidell JC, de Vet HC, Wierdsma NJ, van Bokhorst-de van der Schueren MA. Development and validation of a hospital screening tool for malnutrition: the short nutritional assessment questionnaire (SNAQ). *Clinical nutrition (Edinburgh, Scotland)*. 2005;24(1):75-82.
 17. Likert. A technique for the measurement of attitudes. In: *Archives of Psychology*, 140, 1–55.1932.
 18. Buurman BM, Parlevliet JL, van Deelen BA, de Haan RJ, de Rooij SE. A randomised clinical trial on a comprehensive geriatric assessment and intensive home follow-up after hospital discharge: the Transitional Care Bridge. *BMC health services research*. 2010;10:296.
2. A unique point of intervention that is not duplicated by other approaches.

Author response:

In the ACGH we are working on an evidence-based approach for right care at the right place, and built capacity for acute care within the community and nursing home setting. The components come from other care models for acute care for older persons, such as comprehensive geriatric assessment, the importance of rehabilitation- if possible patients receive physiotherapy twice a day-, function-focused care and transitional care –such as warm handovers. The unique feature is that the facility is located in a facility where all sorts of short-

term and long-term care for older persons are located. Patients have a single room in a home-like environment, that is distinctly different from a hospital environment. Moreover, there is a direct referral from the emergency department (ED) and the care is provided by the same geriatricians working at the ED and hospital.

Please also see figure 1 attached to the revised manuscript, which displays the different components of the AGCH intervention.

Based on your remark we have made a revision the introduction, page 7 lines 159-161:

Lines 159-161:

“This concept of care is new to the Netherlands, to our knowledge there is only one example in Europe to which it compares: a “subacute care unit” in intermediate care, which has been implemented in Spain²¹.”

Response to Reviewer 2

1. The authors present the study protocol for an important project – and an ambitious undertaking – evaluating an acute geriatric community hospital for older adults. Overall, the scope of this study may be too ambitious and difficult to accomplish – there are standard metric outcomes (delirium incidence, readmission, etc.) in addition to patient/caregiver outcomes and economic outcomes.

The study design appears robust and well thought out. The introduction is well written. The caveat of course is that the real world is messier and one wonders about some of the logistics. For example, how many patients and their families are amenable to being transferred from an acute care hospital emergency room to an intermediate care facility like this?

Author response:

Thank you for your comments and your remark regarding the feasibility of patient admission to this geriatric community hospital. In practice our clinical team has experienced that it has been feasible to admit patients to the AGCH. Since its' opening in July 2018 the AGCH has had over 550 admissions. Beyond the scope of this paper: in the past year we have seen that about one quarter of the older adults who required hospitalization were amendable to being transferred from the emergency room to the AGCH.

Between February 1st and December 20th 2019 we were able to recruit 212 patients from the 336 admissions (excluding readmitted study participants) that took place in this period. Please see the patient recruitment diagram in the revised paper, see page 26.

Regarding the different measures that we use, it has been feasible to collect most of the measures, please also see our answer to the remark 1 by reviewer 1 and our answer to your second comment below. We have altered the duration of the measurement of one of the secondary outcomes to improve the feasibility of measuring this outcome, this is the occurrence of delirium. We now only record the measurement of the CAM (Confusion Assessment Method) at day 1, 2 and 3. These measurements are then continued on clinical indication, meaning that they are continued if a patients still show signs of (possible) delirium.

Please also see lines 288-292 in the revised manuscript:

“Nurse practitioners will score the CAM daily from day one till day three of admission, if there are signs of possible delirium at day 3, these measurements are continued until the symptoms are resolved. In addition, during the first three days of admission the Delirium Observation Screening Scale (DOSS) is scored during each nursing shift and is continued when there is a clinical suspicion of delirium.¹⁰”

2. How many patients undergo all the assessment delineated – including quality of life, nutrition, caregiver burden assessment?

Author response:

Thank you for your remark. The 212 participants underwent almost all of the assessments in as described in table 1. However, some patients may not undergo physical tests such as the handgrip strength and gait speed tests as these are part of routine physiotherapy assessment, e.g. patients who are bedbound do not receive. In total, 16 of 212 patients did not wish to fill out a questionnaire, and for this group data were only collected from the medical record. For these participants some of the secondary outcomes (e.g. ADL-functioning) will not be reviewed.

We tried to align most of the data collection with the control cohorts. See the supplementary table on page 4 of this response letter, here we provide an overview of all the assessments and which of these were also conducted in the control cohorts from the Transitional Care Bridge and Hospital-ADL studies.

3. Given that this acute geriatric community hospital has already been in place since 2018, some preliminary/baseline data would be informative.

Author response:

Thank you for your remark. We provide baseline data of the first 212 participants who were recruited between February 1st 2019 and December 20th. Please find the table on page 29 of the revised manuscript and please see the newly added section preliminary data on page 18 of the revised manuscript.

4. This reviewer was left wondering how healthcare utilization will be analysed – the authors state only direct costs would be included. Additional information about this would be important and helpful.

Author response:

We have made an addition to the “Economic Evaluation section” to further explain how health care utilization and costs will be analysed, please see the following lines on page 16-17:

Lines 394-398:

“Direct medical cost will only include costs that are funded through the Dutch healthcare system. The evaluation from a societal perspective will include an estimation of the cost of informal care. Costs will be based on the reference prices found in the Dutch Manual for Costing studies and will be set for final year of data collection (2020 of 2021). According to this guideline costs will be discounted at 4% and quality adjusted life years (QALYs) will be discounted at 1,5 %.”²²

5. For the analysis, this reviewer wonders a bit about the historical control groups. Why was the Transitional Care Bridge Study patient population and the Hospital-ADL patient population chosen to be historical control groups for this current study? Are they truly representative of the general geriatric population or are these patients self-selected or skewed because they are already involved in research studies? This is not a limitation that cannot be overcome, but the authors should discuss this in some more detail in the Discussion. And the justification of choosing these two groups as historical controls could be more fully explained in the methodology.

Author response:

To answer your question regarding why these two populations were chosen as historical control groups: the target population for the AGCH is older persons with acute care needs that need medical treatment, but no diagnostics. Therefore, we hypothesized that we would mainly admit patients with ambulatory sensitive conditions such as infections, exacerbations of COPD or heart failure and frail geriatric patients. In our two historical cohorts patients came from internal medicine, cardiology, pulmonology and geriatrics departments; we know that patients from

these departments have similar diagnosis to those who can be admitted to the AGCH. Moreover we have selected these groups as control groups as these patients come from the same area as the studied population admitted to the AGCH, namely the larger Amsterdam area/region. Secondly, data from these control groups are available to us, as these data were collected by researchers from our research group and clinical department. Thirdly, as this is a non-randomized study, we could also have chosen to set up a prospective control group running alongside the cohort at the AGCH. Unfortunately we did not have funds available for this. Taking these considerations into account we chose to select the Transitional Care Bridge Study and the Hospital-ADL study as control groups. These studies complement each other regarding the measurements that were conducted. Please also see our comment to remark 1 of reviewer 1 and the supplementary table that shows the different measurements that were conducted in these studies.

To answer your question regarding the representation of the general geriatric population and possible skewedness and self-selection in this group: the control cohorts both had slightly different inclusion criteria but represent a general geriatric population of 65 years and older. For the detailed methodology and in- and exclusion criteria of these studies we refer to the published study protocols and papers of these two studies.^{1, 2, 23, 24}

The studies complement each other regarding the patients that were selected: the Transitional care bridge study recruited patients that have both no, moderate or severe cognitive impairment (Mini Mental state exam score <15, via a proxy interview and informed consent and the Hospital-ADL study does not recruit patient with severe cognitive impairment, but has been conducted more recently (2015-2017).

Please also find the additional lines that we have written in the methodology section:

Lines 225-227:

“Two completed studies conducted by our research group were selected as historical control groups. These control groups were selected based on characteristics of the participants - primary admission diagnosis, department, area of residence- and the availability and reproductively of the data.”

And Lines: 239-240

“For the detailed methodology and inclusion criteria of the two control cohorts we refer to the study protocols and papers of these studies.^{1, 2, 23, 24}”

And the discussion section:

Lines 453- 460:

“Our preliminary results show that data collection at the AGCH is feasible and we expect to recruit enough patients to evaluate the primary outcome. There are also limitations to the design of this study. It is a non-randomized study and that historic cohorts are used as control groups. Therefore baseline differences between intervention and control groups may hamper the matching between the groups. Also, as the data from the cohorts were not collected in the same time period as the AGCH cohort there may be external non-observed differences in the Dutch healthcare system and work processes in hospitals may have changed over the years. However, the two control populations were not self-selected and do represent a geriatric population suffering from common exacerbations of chronic conditions and acute illness that occur in older persons.”

6. Regarding the time points, did the authors consider expanding the timeframe to 6-12 months after hospitalization? There may be valuable information at the 12-month (or at least 9 month) time point.

Author response:

We did consider this, however, we think the greatest effects on health outcomes are to be expected soon after the admission to the AGCH, this is why we chose the time points at 1, 3 and 6 months. Secondly, these time points overlap with the time points that were set in the historic control groups. We agree that it would be relevant to know what the long-term outcomes are, such as after 9 or 12 months. In this particular phase of implementation we will

first do a clinical and economic evaluation of this new concept in the Netherlands. In this setting is feasible to collect data at 1,3 and 6 months after discharge. When this study has been completed and there is a proof of concept we could in the future consider a longer timeline and collect follow-up data at 6-12 months post-discharge.

7. Pg. 15: Very robust assessment of patients! This reviewer wonders if all this data collection is feasible in the real world setting (it is a geriatrician's dream wealth of data!)

Author response:

This is a very relevant remark as we are collecting many different parameters in this study. In the design of the study we closely collaborated with the nursing team at AGCH and we have tried to copy data/measurements that are performed by nurses as part of the geriatric assessment of each patient. All the assessments in table 1, page 27 of the manuscript that are performed by doctors (D) nurses (N) and physiotherapists (P) are collected through chart review and are part of routine (geriatric) assessments. We regularly assess feasibility and completeness of data collection with the research team and review this together with clinical staff at the AGCH to reduce missing data. However, as this is a real world setting you are right that there will be missing data. We will analyse this missingness as described on pg. 15-16 of the manuscript.

Regarding the feasibility of recruitment of patients, please also see a flow-chart of the recruitment of patients as can be found on page 26 of the revised manuscript.

8. Similar to the point above re: Pg. 15: On pg. 19, Line 399 the discussion of Process Evaluation – adherence, barriers and facilitators to implementation could be fleshed out in more detail.

Author response:

We have changed this paragraph to concern all the qualitative research that we will conduct. The adherence to the intervention will be part of the quantitative part of the research, e.g. for example when we measure the time between discharge and sending the discharge letter to the general practitioner. Please find the rewritten paragraph "Process evaluation and patient experience" on page 17 of the revised manuscript.

9. I do not find Box 1 on Pg. 25 to be very informative. A figure would be more useful, perhaps showing how patients are recruited in ED → go to the Acute Geriatric Hospital → warm handoff with community nurses → follow up with general practitioners in the community

Author response:

We have replaced box 1 by figure 1 displaying the 'Patient admission process and components of AGCH intervention and goals. Please find this figure as an attachment to the revised manuscript.

10. Pg. 8, Line 124: Change "specialist" to "acute" or "tertiary" care?

Author response:

We have changed this to "secondary" which is in line with the organization of the Dutch Healthcare system, please see page 6 line 136.

11. Pg. 13, Line 264: "This will" is missing a "be"

Author response:

We have added this to the sentence, please see page 11, line 277.

References

1. Buurman, BM, Parlevliet, JL, van Deelen, BA, et al. A randomised clinical trial on a comprehensive geriatric assessment and intensive home follow-up after hospital discharge: the Transitional Care Bridge. *BMC health services research* 2010;10:296.
2. Reichardt, LA, Aarden, JJ, van Seben, R, et al. Unravelling the potential mechanisms behind hospitalization-associated disability in older patients; the Hospital-Associated Disability and impact on daily Life (Hospital-ADL) cohort study protocol. *BMC geriatrics* 2016;16(1):59.
3. Hochberg, Y. A sharper Bonferroni procedure of multiple tests of significance 4. 75 ed. *Biometrika* 1988:800-802.
4. Benjamini Y., HY. Controlling the false discovery rate: a practical and powerful approach to multiple testing. *Journal of the Royal Statistical Society* 1995;57(1):289-300.
5. Charlson, ME, Pompei, P, Ales, KL, et al. A new method of classifying prognostic comorbidity in longitudinal studies: development and validation. *Journal of chronic diseases* 1987;40(5):373-383.
6. Zorginsituut, N. Veelgestelde vragen: polifarmacie 2017. <https://www.gipdatabank.nl/veelgestelde-vragen/polyfarmacie>. Accessed 8th of January 2019.
7. Folstein, MF, Folstein, SE, McHugh, PR. "Mini-mental state". A practical method for grading the cognitive state of patients for the clinician. *Journal of psychiatric research* 1975;12(3):189-198.
8. Heim, N, van Fenema, EM, Weverling-Rijnsburger, AW, et al. Optimal screening for increased risk for adverse outcomes in hospitalised older adults. *Age and ageing* 2015;44(2):239-244.
9. Inouye, SK, van Dyck, CH, Alessi, CA, et al. Clarifying confusion: the confusion assessment method. A new method for detection of delirium. *Annals of internal medicine* 1990;113(12):941-948.
10. Schuurmans, MJ, Shorridge-Baggett, LM, Duursma, SA. The Delirium Observation Screening Scale: a screening instrument for delirium. *Res Theory Nurs Pract* 2003;17(1):31-50.
11. van der Mast, RC, Vinkers, DJ, Stek, ML, et al. Vascular disease and apathy in old age. The Leiden 85-Plus Study. *International journal of geriatric psychiatry* 2008;23(3):266-271.
12. EuroQol--a new facility for the measurement of health-related quality of life. *Health policy (Amsterdam, Netherlands)* 1990;16(3):199-208.
13. Hoogerduijn, JG, Schuurmans, MJ, Duijnste, MS, et al. A systematic review of predictors and screening instruments to identify older hospitalized patients at risk for functional decline. *Journal of clinical nursing* 2007;16(1):46-57.
14. Katz, S. Assessing self-maintenance: activities of daily living, mobility, and instrumental activities of daily living. *Journal of the American Geriatrics Society* 1983;31(12):721-727.
15. Roberts, HC, Denison, HJ, Martin, HJ, et al. A review of the measurement of grip strength in clinical and epidemiological studies: towards a standardised approach. *Age and ageing* 2011;40(4):423-429.
16. Guralnik, JM, Simonsick, EM, Ferrucci, L, et al. A short physical performance battery assessing lower extremity function: association with self-reported disability and prediction of mortality and nursing home admission. *Journal of gerontology* 1994;49(2):M85-94.
17. McCaffery, M, Beebe, A. Pain: clinical manual for nursing practice. CV Mosby: St Louis 1989.
18. Hwang, SS, Chang, VT, Cogswell, J, et al. Clinical relevance of fatigue levels in cancer patients at a Veterans Administration Medical Center. *Cancer* 2002;94(9):2481-2489.
19. Kruizenga, HM, Seidell, JC, de Vet, HC, et al. Development and validation of a hospital screening tool for malnutrition: the short nutritional assessment questionnaire (SNAQ). *Clinical nutrition (Edinburgh, Scotland)* 2005;24(1):75-82.
20. Likert. A technique for the measurement of attitudes. *Archives of Psychology*, 140, 1-55.; 1932.
21. Inzitari, M, Gual, N, Roig, T, et al. Geriatric Screening Tools to Select Older Adults Susceptible for Direct Transfer From the Emergency Department to Subacute Intermediate-Care Hospitalization. *Journal of the American Medical Directors Association* 2015;16(10):837-841.
22. Hakkaart-van Roijen L, V-d-LN, Bouwmans C, Kanters C, Tan SS. Dutch manual for costing in economic evaluations. Diemen National Health Care Institute 2015.

23. Buurman, BM, Parlevliet, JL, Allore, HG, et al. Comprehensive Geriatric Assessment and Transitional Care in Acutely Hospitalized Patients: The Transitional Care Bridge Randomized Clinical Trial. JAMA internal medicine 2016;176(3):302-309.
24. van Seben, R, Reichardt, LA, Aarden, J, et al. Geriatric syndromes from admission to 3 months post-discharge and their association with recovery. Innovation in Aging 2017;1(suppl_1):902-903.

VERSION 2 – REVIEW

REVIEWER	Miharu Nakanishi Tokyo Metropolitan Institute of Medical Science, Japan
REVIEW RETURNED	02-Feb-2020

GENERAL COMMENTS	Thank you for your revisions in response to our previous comments. I found all my comments were adequately addressed in the revision. I have no further comments.
---

REVIEWER	Tammy Hshieh Brigham and Women's Hospital, Boston, USA
REVIEW RETURNED	04-Feb-2020

GENERAL COMMENTS	Some additional English language review and grammar check may be helpful. The authors have been responsive to my comments. * I particularly like the new Supplementary Table. However, I wonder if the Hospital-ADL study and the time points H1 and H2 could have slightly different abbreviations because it can be a bit confusing initially, reading the Table. Maybe keep H1 and H2 but change H-ADL from H to A? * The way delirium is measured using CAM for d1-3 and DOSS each shift while there is concern for delirium is good. * Line 225-227 explaining why these historical control groups were used. The sentence that ultimately was included in the text is not as good as the authors' actual response to reviewers - which highlights the similarities between the three groups. Can the authors incorporate more of their response into the text?
--

VERSION 2 – AUTHOR RESPONSE

Response to Reviewer 2

1. Some additional English language review and grammar check may be helpful.

Author response:

Thank you for this remark, we have done a language check and have made several changes to the manuscript, please see these changes in the manuscript (in yellow).

2. I particularly like the new Supplementary Table. However, I wonder if the Hospital-ADL study and the time points H1 and H2 could have slightly different abbreviations because it can be a bit confusing initially, reading the Table. Maybe keep H1 and H2 but change H-ADL from H to A?

Author response:

We have added the supplementary table that was in the response letter as a separate supplementary table to the submission. In addition we changed all the H (H-ADL) to A in this table.

3. The way delirium is measured using CAM for d1-3 and DOSS each shift while there is concern for delirium is good.

Author response:

Thank you for this remark.

4. Line 225-227 explaining why these historical control groups were used. The sentence that ultimately was included in the text is not as good as the authors' actual response to reviewers - which highlights the similarities between the three groups. Can the authors incorporate more of their response into the text?

Author response:

We have added more details to lines 227-234 and by this have tried to highlight the similarities between the three cohorts:

"We selected two completed cohort studies that were conducted by our research group as historical control groups. We expect that the patients from these cohorts have similar admission diagnoses as those who can be admitted to the AGCH, namely, diagnoses that are ambulatory care sensitive conditions such as infections and exacerbations of COPD or heart failure. Patients in these two cohorts were admitted to internal medicine, cardiology, pulmonology and geriatrics departments. These departments admit patients with diagnoses similar to those that can be admitted to the AGCH. In addition, we have selected these cohorts as control groups as the patients come from the same area as the studied population admitted to the AGCH, that is, the greater Amsterdam area."

Please see the manuscript for minor changes to the rest of the paragraph.